# Trail Conditions and Community Use: Utilizing Geospatial Video to Guide the Adoption of a Spatial-Temporal Trail Audit Tool (STAT)

**DOI:** 10.3390/ijerph18168741

**Published:** 2021-08-19

**Authors:** Carissa Smock, Naomi Carlson, Chelsey Kirkland

**Affiliations:** 1School of Business, Northcentral University, La Jolla, CA 92037, USA; 2Department of Active Transportation, Headwaters Regional Development Commission, Bemidji, MN 56601, USA; ncarlson@hrdc.org; 3College of Public Health, Kent State University, Kent, OH 44240, USA; cbruce5@kent.edu

**Keywords:** physical activity, socio-ecological model, geospatial video, spatial-temporal, trail audit tool, rural

## Abstract

Physical activity (PA), associated with all-cause mortality, morbidity, and healthcare costs, improves vitamin D absorption, immune response, and stress when completed outdoors. Rural communities, which experience PA inequities, rely on trails to meet PA guidelines. However, current trail audit methods could be more efficient and accurate, which geospatial video may support. Therefore, the study purpose was (1) to identify and adopt validated instruments for trail audit evaluations using geospatial video and a composite score and (2) to determine if geospatial video and a composite score motivate (influence the decision to use) specific trail selection among current trail users. Phase 1 used a mixed-method exploratory sequential core design using qualitative data, then quantitative data for the development of the Spatial-temporal Trail Audit Tool (STAT). Geospatial videos of two Northeast Ohio trails were collected using a bicycle-mounted spatial video camera and video analysis software. The creation of STAT was integrated from Neighborhood Environment Walkability Scale (NEWS), Walk Score, and Path Environment Audit Tool (PEAT) audit tools based on four constructs: trail accessibility, conditions, amenities, and safety. Scoring was determined by three independent reviewers. Phase 2 included a mixed-method convergent core design to test the applicability of STAT for trail participant motivation. STAT has 20 items in 4 content areas computing a composite score and was found to increase trail quality and motivation for use. STAT can evaluate trails for PA using geospatial video and a composite score which may spur PA through increased motivation to select and use trails.

## 1. Introduction

### 1.1. Importance of Physical Activity

Physical activity (PA) is associated with a reduction in all-cause mortality, morbidity, and associated healthcare costs [1,2]. Physical inactivity is a public health problem globally as 1 in 4 adults of adults aged 18 years and older do not meet the globally recommended levels of PA [3]. These global recommendations of PA include 150 min of PA per week [3]. There is also a dose-response relationship, as adults who attain a minimum of 150 min of moderate-intensity or 75 min of vigorous-intensity exercise per week demonstrate a 15% mortality reduction across the lifespan, whereas those who accrue at a minimum of 300 min per week demonstrate a 26% mortality reduction [4]. While adult PA has significantly increased in the past decade (from about 18% of adults in 2008 to about 24% of adults in 2018) [5], about 75% of adults are still not meeting the above guidelines [6], demonstrating the need for PA interventions [7].

### 1.2. Inequities for PA in Rural Environments and the Socio-Ecological Model

A large number of adults do not meet PA recommendations due to inequities in access to safe environments for PA [7]. Some of these inequities are found in rural environments where only 19.6% of people are meeting PA guidelines compared to 25.3% among those living in urban communities [8,9]. The Spatial-temporal Trail Audit Tool (STAT) was developed using trails that run through a mix of rural and urban areas as defined by the proposed, updated United States Census Bureau area designations (an area fewer than 4000 housing units or a population less than 10,000) [10]. While STAT can be used on any trail, it is especially helpful in rural areas challenged with vast space and limited resources [11]. Rural women’s PA rates are even lower than men’s and especially those with low income [12]. Furthermore, rural communities have higher obesity rates, injury, and fatality rates from collisions, and poorer PA infrastructure, which have worsened during the COVID-19 pandemic [8].

The socio-ecological model (SEM) includes domains that impact PA, including individual, social/intrapersonal, community/environment (built and natural), and policy, needed to address complex challenges in rural communities. This better enables the promotion of PA across all the socio-ecological domains as having accessible, highly rated trails is shown to increase PA in rural populations [10,11,13]. Specifically, at the individual level, people in rural populations are more likely to access locations for PA that are free (such as a trail or neighborhood streets) as well as locations that can be easily accessed with others intersecting at the social/intrapersonal level [14]. At the community/environment level, the use of trails is associated with meeting PA recommendations in rural populations [15]. Finally, at the policy level, work needs to be conducted to ensure quality and safe conditions for trail use. Therefore, increased surveillance of rural trails for PA intersects multiple SEM domains and is needed to determine quality and usage patterns.

### 1.3. Importance of Trails for PA

In addition to increased access and use, engaging in trail PA provides the additional benefits of being outdoors. These benefits include vitamin D absorption and additional immune response and stress reduction [16,17]. The associations between trail conditions and community use illustrate the importance of testing increased PA uptake through trail audits [18]. At the same time, PA interventions addressing trail access inequities currently lack efficient and accurate trail audit methods. Trail audits include ways for researchers and practitioners to identify trail quality and utilize various measures and methods. Trail audit tools are designed to measure trail use facilitators.

### 1.4. Need for Increased Accuracy and Efficiency in Trail Quality and Use Measures and Methods

Trail quality is currently assessed through audit tools designed to measure trail use facilitators and barriers. Facilitators include groomed trail conditions, adequate lighting, mixed views (urban and natural scenery), and trailside amenities (e.g., restrooms, benches) [18,19]. Barriers include excessive noise, debris, dense vegetation, tunnels, non-visual drainage feature, and geographic factors (e.g., safety and accessibility) [18,19]. Trail audits can be particularly difficult to collect in rural communities challenged with vast space and limited resources [11]. Currently, trail use data collection methods include manually counting trail users with clickers or infrared video counters [20]. The former is time-intensive, and the latter has inaccuracies due to weather and wildlife interferences and cannot distinguish unique users [20]. Therefore, more efficient and accurate trail quality and use data collection methods are needed for practitioners and researchers focused on improving the current trail conditions and use.

### 1.5. Geospatial Video

Geospatial video may assist with trail quality measures and trail use collection methods as it captures both quality and use of PA access points, such as parks and trails, by combining Global Positioning Systems (GPS) with video. Mills et al. (2010) [21] developed this methodology, which enables data collection efficiency, the ability to survey locations over multiple time periods, and generate archival data, thus, places can be revisited through the video. This approach (1) increases field data collection efficiency while (2) increasing accuracy. Previously geospatial video has been used in disaster recovery efforts as well as surveillance of populations who are homeless [21,22].

Trail quality and use data can be collected more efficiently by attaching the camera to a bicycle, rather than walking the trails, and archived to allow multiple researchers or practitioners to audit locations over multiple time periods [21]. This results in the ability to archive trail conditions and be used to determine time and place relationships, such as specific trail features, resulting in increased use during specific times of the day or year and increases validity and reliability of the trail audit. Additionally, geospatial video provides researchers with a way of using this mechanism to include qualitative data into audit tools [21], providing significant improvements on the previous methodology. Furthermore, geospatial video can capture trail accessibility, conditions, amenities, and safety, supporting intuitive dissemination amongst colleagues and stakeholders. Geospatial video has not, to our knowledge, been used to measure trail quality and use, resulting in a need for validation.

### 1.6. Study Purpose

Current trail audit methods could be more efficient, accurate, and especially helpful in rural areas as having accessible, highly rated trails is shown to increase PA in rural populations. Using geospatial video with a redesigned audit tool may result in better audits and lead to improved PA interventions. Therefore, the purpose of this was (1) to identify and adopt validated instruments for trail audit evaluations using geospatial video and a composite score and (2) to determine if geospatial video and a composite score motivate (influence the decision to use) specific trail selection among physically active adults.

## 2. Materials and Methods

To create and validate the final quantitative trail quality and use instrument, STAT, this study was designed as a 2-phase mixed-methods study. Please see Figure 1 for a study design pictorial representation. Capital letters within Figure 1 represent which data type, quantitative or qualitative, was the prioritized method, and lowercase letters are used to represent the less focused on data type in mixed-methods [23]. Phase 1 used Creswell and Plano Clark’s (2018) [23] mixed-methods exploratory sequential core design using qualitative data then quantitative data for the purpose of development of STAT [24]. Phase 2 used Creswell and Plano Clark’s (2018) [23] mixed-methods convergent core design, which served the purpose of triangulation, converging, and corroborating the study findings through comparing the qualitative and quantitative data to test and revise STAT [24].

### 2.1. Phase 1

Phase 1, mixed-methods exploratory sequential core design, consisted of 3 steps. The first step was to collect qualitative data in the form of a geospatial trail video. The second step was to use previously validated quantitative built environment trail audit tools to create STAT. The third step integrated the qualitative and quantitative data to revise STAT. The rationale for this approach was for the development of STAT [24], and the qualitative data allowed the researchers to understand the measures and questions needed for the quantitative STAT [23].

#### 2.1.1. Step 1: Geospatial Video Collection

As seen in Figure 2, portable, bicycle-mounted geospatial video cameras (Contour +2 model 1700) and the coordinating video analysis software (Contour Storyteller) were used to acquire, analyze, and test video of the 2 Northeast Ohio trails: West branch trail and Portage hike and Bike trail (Figure 3). These trails were selected based on researcher convenience and trail usage. The West branch trail is considered the most frequented free PA access point in Canton, Ohio, and has a broad range of users [25]. The Portage hike and Bike trail, centrally-located and spanning nearly 10 miles, connects Kent and Ravenna, Ohio, and is also within walking distance from Kent State University’s Kent campus. Trails included a mix of rural and urban areas [10,26]. Video of the trails was collected across 6 data collection outings, 3 for each of the 2 trails. The West branch trail data were collected by author C.S., and the and Portage hike and Bike trail data were collected by author N.C. All videos were collected at 4 pm Eastern Standard Time weekly as trail use may vary around users’ schedules as well as around seasonal and daily temperatures and daylight [27]. Each ride began from the same starting point and in the same direction (south to north). The resulting videos from these rides were uploaded into Contour Storyteller, the camera’s accompanying video software, for further analysis.

#### 2.1.2. Step 2: STAT Creation

To create STAT, content analysis of validated trail audit tools was conducted by adopting previous applications of geospatial video [21,28]. Three independent reviewers examined the trail videos to determine previously-validated built environment audit tools’ content that could be scored using video content. The audit tools from which items and constructs were adopted included the Neighborhood Environment Walkability Scale (NEWS), Walk Score, and Path Environment Audit Tool (PEAT).

NEWS, developed in 2002 by Dr. Sallis, measures residents’ perception of neighborhood design features (environmental attributes), related to PA, within their local area [29]. NEWS addresses constructs of residential density, land use mix (including both indices of proximity and accessibility), street connectivity, infrastructure for walking/cycling, neighborhood aesthetics, traffic and crime safety, and neighborhood satisfaction [30]. NEWS has been well-documented, tested, and validated in multiple countries, and additional modifications have been made to abbreviate the tool to various situations [29]. We further adopted NEWS to specifically measure these constructs within trails.

Walk Score, developed in 2007 by Herstas, company Chief Executive Officer [31], is a previously-validated built environment audit tool that links PA and proximity to one’s destination, utilizes Geographic Information Systems (GIS) to measure and estimate neighborhood walkability via the physical distance to 13 specific destinations, including: grocery stores, parks, book stores, coffee shops, restaurants, bars, movie theaters, schools, libraries, fitness centers, drug stores, hardware stores, and clothing/music stores [32]. Walk Score is applicable across a variety of geographic settings and spatial scales [33]. Similarly, to NEWS, we further adopted Walk Score to specifically measure these constructs within trails.

Finally, PEAT was developed in 2006 by Troped, Cromley, Fragala, Melly, Hasbrouck, Gortmaker, and Brownson to assess the physical characteristics of community trails and paths, including their design, amenities, and aesthetics [34]. PEAT has displayed inter-rater reliability and validity for the majority of its items [34]. We added those constructs identified as important to trail users from NEWS and Walk Score not included in PEAT.

Beginning with validated question content from the original NEWS survey and thereafter incorporating elements of Walk Score and PEAT, researcher M.N. and authors N.C. and C.S. compiled a list of content relevant to physical trail conditions from which the language of these selected questions was later modified to address these concerns specifically. After an initial list was outlined, N.C., M.N. and C.S. then reviewed the spatial video and added additional question content evoked from the viewing. Each reviewer’s questions were again compiled and organized into categories, which were based upon the existing built environment tool categories and question themes that occurred naturally.

STAT composite scores were calculated from 0 to 100 to produce a general trail composite score by adding together the 4 content areas of access, conditions, amenities, and safety. Within the content areas, there were 20 items that used a 0 to 5, 6-point Likert Scale in which 0 equals “not at all” and 5, “yes, very much”. An additional composite score was produced for the low- and high-intensity PA subsets, ranging from 0 to 25, by adding together scores from items using a 0 to 5, 6-point Likert Scale as well.

#### 2.1.3. Step 3: STAT Revision

Following the initial audit tool adoption and construction, STAT was reviewed by two experts in built environment research and one in research design and survey methods, respectively, to provide face validity and identify potential content gaps. Following this input, the researchers integrated the qualitative and quantitative data by reviewing the trail geospatial video, using STAT to conduct an audit of the trails using the video, and revised STAT based on the experts’ feedback and trail audit.

### 2.2. Phase 2

Institutional Review Board approval was obtained for human subjects research from the first author’s university (protocol code 16-597). Phase 2, convergent core design, consisted of steps 4 and 5 in the study. Step 4 simultaneously collected qualitative (focus group) and quantitative (online survey) data about the applicability and usability of STAT. Step 5 integrated and compared the qualitative and quantitative data to revise STAT. The rationale for this approach is that this allowed the researchers to compare, corroborate, and examine facets of STAT through triangulation and convergence of the data, which developed a more complete picture and tested the applicability and usability of STAT [24,25].

#### 2.2.1. Step 4: Test STAT Focus Group

A convenience sample of participants, including close to equal representation of males and females, was then recruited using email from self-identified physically active participants at the first author’s university to join the focus group. Already physically active adults were recruited because they were familiar with the access, conditions, amenities, and safety of PA locations that tend to increase their selection of specific locations for PA. Therefore, using already active people provides clear insights into how STAT will work, which researchers and practitioners can use to improve the trails and increase the probability that those not meeting physical activity guidelines will be encouraged to use the trails. A semi-structured focus group was chosen to elicit participants’ thoughts and perceptions of STAT through intra- and inter-personal conversations for more open dialog supplementing the STAT survey in step 4 [35]. Members of local area running groups’ message boards were recruited to complete a survey. These self-identified cyclists, walkers, and runners tested the applicability of the geospatial techniques to collect data using the validated tools described to provide perceptions of the importance of the video, individual items, constructs, and composite score of STAT. We also assessed trail user motivation that examined which components of STAT influenced participants’ decisions to select a specific trail.

Participants were given a copy of the STAT items and constructs, asked to review and then share which items and constructs were important for selecting a trail for exercise, items they would change or add to help them select a trail, and what was most important in selecting a trail for exercise. Next, participants were asked to watch a trail video played for them at the focus group. After watching the video, participants were asked if the video, specific categories of items, or a combination would help them determine whether or not they would select a trail for exercise and if having any of this information would motivate them to do so. Finally, participants were asked about their trail use and PA. The focus group discussion was transcribed, and focus group participants were asked to record notes on the copy of the STAT items, which included a copy of the focus group questions to assist in validating responses.

After all focus groups were completed, researcher M.N., and authors N.C. and C.S. transcribed and de-identified files. The transcripts were entered into QSR International’s NVivo 11 qualitative data analysis software and analyzed using Saldaña’s (2016) [36] evaluation and descriptive coding. Evaluation coding includes applying codes to qualitative data to assign judgment about the merit, worth, or significance of trail quality and use measures. Descriptive coding includes labeling assigned to data to summarize in a word or short phrase. The basic topic of the qualitative data passage provides an inventory of topics for indexing and categorizing.

#### 2.2.2. Step 4: Test STAT Survey

The survey included STAT items and participants were asked to rate each item for importance of selecting a trail for exercise using a Likert scale with 0 being not at all to 5 being very much. Participants were also asked which items they would change or add to help them select a trail, and what item was most important to them when selecting a trail for exercise. Next, participants were asked to watch a trail video linked to the survey. After watching the video participants were asked if the video, specific categories of items, or a combination would help them determine whether they would select a trail for exercise and if having STAT information would motivate them to do so, again, using a Likert scale with 0 being not at all to 5 being very much. Finally, participants were asked about their trail use PA, using the valid and reliable International Physical Activity Questionnaire (IPAQ) [37,38], and demographics including race, age, gender, and income. All survey data were analyzed with descriptive statistics using IBM SPSS Statistics for Windows, Version 21.0. This supplemented the qualitative data from focus groups addressing the second part of the research purpose, to determine if geospatial video and a composite score motivate (influence the decision to use) specific trail selection among physically active adults.

#### 2.2.3. Step 5: Final STAT Revisions

The final STAT was created from the integration of the qualitative and quantitative data collected in step 4. This integration was completed by analyzing patterns between focus group and survey data to compare findings from the qualitative and quantitative data [34]. Data were assessed using parallel constructs for both types of data and results were compared through transforming the qualitative data set into quantitative scores through ranking and jointly displaying both forms of data [23]. The two types of data provide validation for each other and also create a foundation for deriving decisions about changes to the items in the STAT tool while providing triangulation for perceptions about whether or not the video, specific categories of items, or a combination would help them determine if they would select a trail for exercise and if having any of this information would motivate them to do so.

## 3. Results

### 3.1. Step 2 Content Adopted from Validated Instruments

After reviewing NEWS, Walk Score, and PEAT audit tools, the researchers pulled content areas from these tools that served as a valid, theoretical basis including four constructs: (1) trail accessibility, (2) trail conditions, (3) trail amenities, and (4) trail safety. Constructs from the three validated instruments that could not be captured using the video were excluded from survey construction. These include (1) types of neighborhood residences; (2) stores, facilities, and other neighborhood amenities; and (3) neighborhood satisfaction. Constructs adopted and modified include (4) safety from traffic, and (5) safety from crime; the latter serving as a basis for STAT’s trail safety content. Please see Appendix A, Table A1 for the final STAT.

These validated instruments’ constructs were adopted based on what the researchers could capture using the video from which STAT was created and comprised of 20-items measuring trail quality. Trail audit tools also cannot archive or share collected data. Therefore, in conjunction with the adopted audit tool (STAT), a video-based composite score was calculated using geospatial techniques. The researchers determined that a geospatially enhanced instrument, combining items from three validated built environment audit tools, was most appropriate based on the ability to measure the items using geospatial techniques to archive and therefore share the collected data.

A composite score was calculated by adding together items from 6-point Likert scales. The scales included a not applicable inclusion (0), allowing for greater generalizability without devaluation of the composite score since (1) as the lowest option assigns value to a score that should have no value. An additional composite score was produced for the low- and high-intensity PA subsets, ranging from 0 to 25.

### 3.2. Step 3 STAT Test Results

During step 3, STAT was tested using the geospatial video collected in step 1. While researcher M.N. and authors N.C. and C.S. had different scores from each other (low interrater reliability), every score was within two points of each other, and consensus of the scores between the researcher and authors was determined and are reported in Table 1.

### 3.3. Step 4 Focus Group Responses

Participants from the focus group (*n* = 30) were an average age of 44, mostly White, 53% male, $70,000 average income, and averaged about 70 min per trail session 4.2 days a week. Participants also indicated that the type of exercise they planned to do informed which trail they would select. Please see Table 2 for survey participant demographics.

The focus group analysis resulted in six themes. The frequency of these themes as well as participant ranking indicated that safety was most important in selecting a trail for exercise. Trail amenities, conditions, and access were also highly important, aligning with the content areas included in STAT (Table 3). Safety was described by a participant as, “…*how much a trail interacts with automobile traffic. Specifically, how much time must I spend riding in the street. In winter, if the trail is clear of snow and ice*”. Another participant shared that amenities should include “…*good trail markings and mileage markers, restrooms and water refill stations accessible*”. A third participant shared the importance of trail of conditions, “*I would prefer the level of muddiness on trail sections to be low when possible*”. In terms of access, one participant shared, “…*for some folk, I am assuming that wheelchair or stroller capabilities are important. I prefer my trails a bit raw less groomed*”.

Participants shared that exercise type determined their trail selection. Therefore, they would only be motivated to use a new trail if it was appropriate for and supported the activity they were planning to do. A participant shared that this included “*…length of the trail and ability to make your running route longer/shorter (multiple loops available)*”. The motivation was increased if “…*the trail has elevation change, is at least somewhat technical*”. Lastly, participants shared that they would use the video, amenity scoring, and condition items to help them determine a trail was appropriate for and supported the activity they were planning to do if they had access to the trail.

### 3.4. Step 4 Survey Responses

Participants from message boards (*n* = 56) were slightly older than focus group participants with an average age of 46, similar to focus group participants in that they were mostly White and 55% male, with a higher average income of $74,000, and averaged more time, about 100 min per trail session, but less (2.3) days a week. Similar to the focus group participants, survey participants also indicated that the type of exercise they planned to do informed which trail they would select. Please see Table 4 for survey participant demographics.

The importance of STAT items for survey participants is in Table 5. Survey participants indicated that STAT items most important to them were how long the trail segments were followed by if there was any loitering or suspicious activity. The least important items included if there were picnic areas followed by if it was well lit and traffic speed.

As indicated by the mean score of the Likert scales, survey participants indicated that using the STAT composite score alongside the video to select a trail would be slightly more helpful in assisting them in planning to use a trail and in motivating them to use a trail than one or the other”. Trail conditions are slightly more important to survey participants than the other constructs, followed by safety, amenities, and access (Table 6).

When comparing focus group and survey data, both participant groups indicated that exercise type determined their trail selection rather than the opposite. Seeing a new trail would only motivate them to select it if it was appropriate for and supported the PA activity they were planning to do (Table 7). Table 7 shows the ranking comparison of how focus group and survey participants viewed the importance of STAT constructs. While focus group participants indicated safety was the most important construct, survey responses indicated access was the most important construct. The only category that ranked the same between the groups was conditions as the third most important construct.

## 4. Discussion

The study purpose was (1) to identify and adopt validated instruments for trail audit evaluations using geospatial video and a composite score and (2) to determine if geospatial video and a composite score motivate (influence the decision to use) specific trail selection among physically active adults. Current methods of measuring the use of trails for PA includes counting users, as well as noting types of PA at trail heads or using infrared cameras, which are often ineffective and inaccurate. Therefore, using a bicycle-mounted camera to collect trail quality and use simultaneously is an effective, time-saving approach to data collection, enabling access to locations not reachable by car and providing a faster alternative to walking. An analysis of male versus female results was not completed since equal representation was ensured, and it was outside the scope of this research purpose. Outcomes of STAT can be used by a variety of stakeholders to increase trail quality and subsequent trail use.

### 4.1. STAT Uses

Geospatial methods and trail audits (such as STAT) have numerous uses. First, trail user motivation to select and use a trail based on video and the composite score would be strengthened. This may be helpful for researchers, practitioners, employee health coordinators, health insurance companies, and clinicians interested in health promotion efforts focusing on access to locations for exercise. Outdoor spaces may offer a more sustainable solution than costs and commute time associated with a traditional gym membership. Furthermore, safety and accessibility measures included in STAT should be considered alongside familiarity, distance to the location, physical ability, and socioeconomic access when referring patients to locations for PA [39]. Thus, strengthening the healthcare system and public land links, including local trails, can help medical communities promote healthy, lower-cost activities that can be incorporated into the patients’ daily life.

Another potential use for STAT is to monitor trail maintenance for park district paths at all levels (community, state, and national) and community use over time. Neighborhoods could be evaluated for walking, cycling, and running for recreation and active transit, including shared or dedicated bike/pedestrian paths and safe routes to schools, parks, libraries, and other community locations. Grant writers applying to awards to enhance these locations (e.g., parks, schools, neighborhoods, active transit) are often required to include path or trail audit results in their applications. STAT provides both these audit results and examples such as frame images and video links, enhancing grant applications.

Additionally, community safety concerns could be addressed by matching city crime data with STAT’s trail safety perceptions. By testing trail safety perception with actual city crime data, community concerns can be addressed as needed through communication or police department surveillance as needed. Moreover, trail condition and amenity outcomes can be mapped and distributed to stakeholders and community members to increase the passage of park levies and other trail improvement and upkeep.

### 4.2. Rural Implications

Rural communities may find STAT particularly helpful as they manage more geographical space with fewer resources, including staff. These communities also tend to have poorer health outcomes, including PA [8], making safe access to high quality and trails and paths for PA even more important. Therefore, STAT results could be disseminated to community hospitals, government officials, stakeholders, and community members to raise awareness and address rural trail inequities through financial support such as grants.

The use of trails is associated with meeting PA recommendations in rural populations and at the same time, aside from one intervention, Heartland Moves (HM) [15], study settings do not focus on trail quality or use [40,41,42,43]. Rather, settings tend to include schools, churches, community locations, worksites, medical practices and clinics, participants’ homes, community or recreation centers and are sometimes unreported [40,41,42,43]. Though these studies have mixed results on increasing rural PA, there is some evidence to support the effectiveness of theory and models, including the SEM [13] and community-based interventions that include low to moderate-intensity exercise to increase PA, physical function, and psychological state [43]. Authors recommended that future rural interventions focus on reported values and interests, such as personal relationships and being outdoors, and focus on those rural populations who do not meet PA recommendations [15]. More rigorous studies are required to identify the most critical characteristics of community-based interventions for rural settings, which may be supported through STAT.

### 4.3. Future Research Directions

Although each of the three built environment audit tools (NEWS, Walk Score, PEAT) have been well-documented, tested, and validated, our adopted version of these tools, STAT, should undergo further validation among both diverse samples of trail users and practitioners, grant writers, and researchers. Additionally, while the development of STAT was both inspired by and constructed in conjunction with the use of geospatial video, the use of STAT is not limited to this technology. Geospatial video is, in general, a helpful dissemination tool and communication aid to share trail conditions and results with community stakeholders. While Plano Clark’s (2018) [23] exploratory sequential core design helped create and validate STAT, it requires ensuring participants included in the qualitative strand are different than those in the quantitative strand. Although the authors followed this during recruitment, it could not be ensured that there was no overlap. Therefore, future research should use a large sample of individuals who are very different (e.g., location, age, income) than those included in this research. Due to using trails that run through a mix of rural and urban areas to develop STAT, it should be further tested in rural areas.

While Plano Clark’s (2018) [23] exploratory sequential core design in phase 1 helped create and validate STAT, it requires ensuring participants included in the qualitative strand are different than those in the quantitative strand. Although the authors followed this during recruitment, it could not be ensured that there was no overlap. Therefore, future research should use a large sample of individuals who are very different (e.g., location, age, income) than those included in this research. Phase 2 used Plano Clark’s (2018) [23] convergent design, which recommends using equal quantitative and qualitative sample sizes. The researchers were not able to achieve this, but the intent of the sample sizes was different as the focus group had a smaller sample size to ensure each participant had a chance to speak, and the survey was to obtain group means. Future research could build on this strategy by gathering data to compare quantitative group means with qualitative individual experiences through interviews.

Though geospatial video is instrumental for STAT, it likewise poses noteworthy temporal limitations. Captured geospatial data are physically limited to what was captured during the collection outing, characterized by the day and time of data collection. Since trail condition is variable depending upon the time of day, the time of year, and weather condition, temporality could alter the physical condition and, subsequent evaluation of the trail. As such, trail evaluations utilizing spatial video to administer STAT should collect multiple videos across a variety of times (such as seasonally) and continue routinely to establish archival data. Finally, a central platform for geospatial video storage, retrieval, and dissemination would be helpful due to large file sizes.

## 5. Conclusions

STAT increases efficiency and accuracy of assessing trail quality and use for PA using multiple levels of the SEM compared to current methods of counting users, noting types of PA at trail heads, and infrared cameras. Furthermore, since STAT uses geospatial video, a bicycle-mounted camera, it simultaneously collects trail quality, special temporal relationships and provides access to locations not reachable by car. Additionally, trail users indicated motivation to select and use trails based on STAT’s video and composite score. Therefore, a variety of stakeholders and communities may find STAT useful for increasing trail quality and use. Rural communities may especially benefit from STAT to promote PA across a variety of industries and environments. STAT results could be disseminated to address inequities through potential supports such as grants, levies, and other applications for funds.

## Figures and Tables

**Figure 1 ijerph-18-08741-f001:**
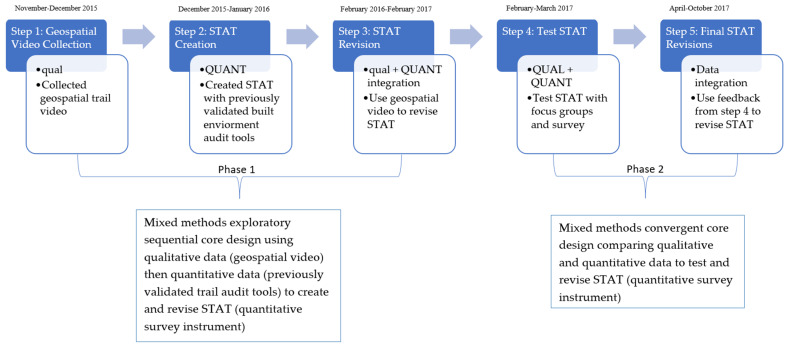
Study timeline and pictorial representation of study design.

**Figure 2 ijerph-18-08741-f002:**
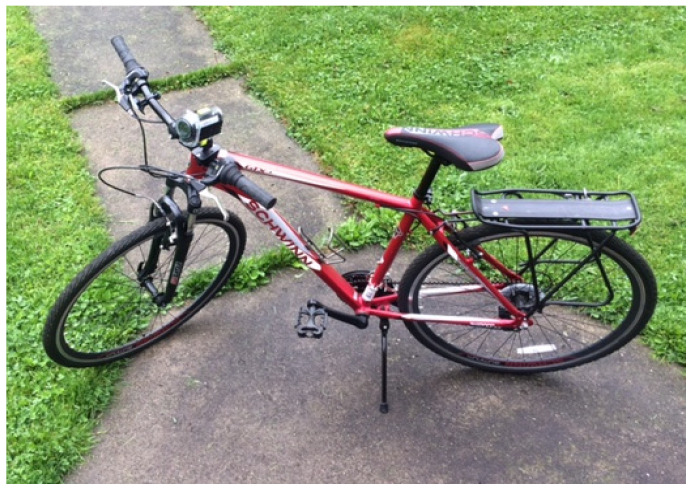
Portable, bicycle-mounted geospatial video camera.

**Figure 3 ijerph-18-08741-f003:**
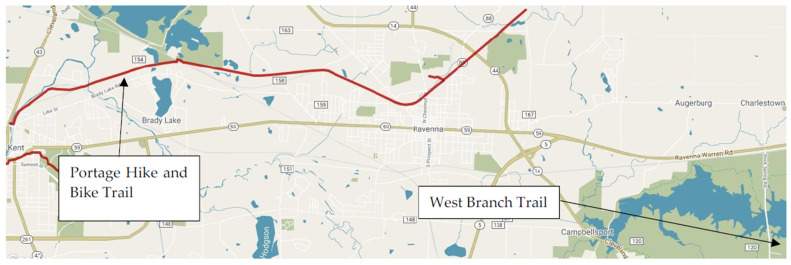
Map of Portage hike and Bike trail and West branch trail. Note. From trails-to-trail conservancy. Portage hike and Bike trail. Available online: https://www.traillink.com/trail-maps/portage-hike-and-bike-trail/ (accessed on 5 August 2021).

**Table 1 ijerph-18-08741-t001:** Scoring of the Ohio trails.

Measure (Max Score)	Trail A: Canton	Trail B: Kent
Access (30)	26	24
Conditions (15)	15	14
Amenities (20)	18	20
Safety (35)	30	32
Composite Score (100)	89	90
Lower intensity exercise (25)	21	23
Higher intensity exercise (25)	19	12

**Table 2 ijerph-18-08741-t002:** Focus group participants.

Measure	*n*	Mean/Percentage *
Age	30	46
Race		
White	28	93.3%
Choose not to answer	2	6.7%
Sex		
Male	16	53.3%
Female	14	46.7%
Income	30	$70,000
Minutes per trail session	30	69.6
Days per week trail session	30	4.2
Trail determines activity	30	100%

* Mean and percentage of frequency are reported.

**Table 3 ijerph-18-08741-t003:** Focus group themes.

Theme	Sample Quote	Frequency
Safety	“*It is important to have no poison ivy, oak on the trails*”.	30
Amenities	“*Water at trailhead. Information board at trailhead with maps and information about connecting trails*”.	26
Conditions	“*Split litter and debris into two different items. No one wants litter, but I don’t care about downed branches, leaves etc*”.	23
Access	“*Proximity of a trailhead that I can actually ride my bike to* vs. *driving*”.	21
Selection decision	“*overall length of trails-high priority*”	19
Motivation	“*Whether the trail can be a challenge*”	12

**Table 4 ijerph-18-08741-t004:** Survey participants.

Measure	*n*	Mean/Percentage *
Age	38	46
Race		
White	35	92.1%
Choose not to answer	2	5.3%
Asian/Pacific Islander	1	2.6%
Sex		
Male	21	55.3%
Female	17	44.7%
Income	35	$74,000
Minutes per trail session	38	100.8
Days per week trail session	35	2.3
Trail determines activity	37	91.9%

* Mean and percentage of frequency are reported.

**Table 5 ijerph-18-08741-t005:** Importance of STAT items for survey participants (*n* = 56).

Construct	Measure	Mean/Percentage *
Access	Parking	3.93
	Connecting	3.79
	Cyclists/Pedestrians	2.54
	4-way intersections	2.62
	Signage	3.80
	Segments	4.23
Conditions	Groomed	2.38
	Litter Free	3.77
	Short Intersections	2.02
Amenities	Bridges	2.80
	Picnic Tables	1.50
	Restrooms	3.02
	Shade	3.25
Safety	Street Separation	3.68
	Traffic Speed	1.96
	Blind Spots	2.09
	Loitering	4.14
	Graffiti	3.29
	Well lit	1.91
	Can see others	2.56

* Mean and percentage of frequency are reported.

**Table 6 ijerph-18-08741-t006:** Use of STAT to select trails (*n* = 35).

Measure	Construct	Mean/Percentage *
Geospatial Video	Planning	2.14
	Motivating	2.26
STAT Constructs	Access	2.63
	Conditions	2.91
	Amenities	2.83
	Safety	2.86
	All constructs used to select trail	2.91
	Geospatial video used to select trail	2.71
	STAT used to select trail	2.94

* Mean and percentage of frequency are reported.

**Table 7 ijerph-18-08741-t007:** Qualitative and quantitative joint table: ranking comparison of focus group and survey scores.

Construct	Focus Group	Survey
Access	4th	1st
Conditions	3rd	3rd
Amenities	2nd	4th
Safety	1st	2nd

## Data Availability

The data are not publicly available due to privacy concerns.

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
