# Peer review of "Trail Conditions and Community Use: Utilizing Geospatial Video to Guide the Adoption of a Spatial-Temporal Trail Audit Tool (STAT)"

_ijerph, 2021, doi:10.3390/ijerph18168741_

Round 1
Reviewer 1 Report
As I understand it, the aim of this paper is to outline the methods used to design and develop STAT, a tool to measure trail conditions and quality for purpose of promoting PA. Please see the comments below as constructive and aimed to improve your manuscript.
INTRODUCTION
Overall, I feel like this section needs to be re-focused to make it clearer what the need for the STAT development is. This section is lengthy, but doesn't provide sufficient synthesis. Specifically:
- It is appropriate to outline the need for PA promotion, i.e. health benefits, but I'm not sure that your specific reference to COVID-19 in this section is particularly necessary, given that it's not the focus of the paper and there are similar benefits for various diseases/conditions.
- This section would benefit from a clearer outline of the population you are speaking to, as you make reference to PA rates and guidelines, but these are not the same globally.
- You make reference to adults, but don't specify this age group.
- You also mention rural, but need to define this (as it varies globally) and make it clear whether STAT was developed rurally or is for use rurally, or on trails in general.
- Be clear about whether you mean 'not making PA guidelines' or 'inactive'.
- I'm not sure whether you need to discuss the SEM here, as it poses little relevance to your study design. If your argument is that we need to be better able to promote PA and having accessible/highly rated trails are part of that, you might reference the SCT, which in this context refers to factors that might enable PA.
- Your study purpose is too vague and you need to better summaries your introduction section to better justify the need for the study.
METHODS
This section is also very long and at times missed some key information, especially around participant recruitment. Specifically:
- Your use of capitals to denote the phase/type of data collection is not consistent and confusing- please consider another way to present this.
- I'm not sure what figure 1 adds to this section- perhaps an adapted figure that explains the timeline might help the reader.
- You should not refer to authors/developers by first name- you do this multiple times.
- Be clearer about why the scales/measures/tools that you have used to develop the STAT are inadequate- to help justify the need for your tool, as this is not clear.
- IRB approval? Is there a reference number?
- How did you come to your convenience sample- this needs to be more clear.
- What does 'adequate representation' mean?
- Why use already active people when your main argument in the introduction is that we need to be able to promote inactive or people not meeting PA guidelines to become active- please explain how your population is therefore representative?
- In terms of your qualitative analysis, can you please provide more information about the method/ology you undertook, and justify why you used this method/ology over another.
- What analyses were done in SPSS- to answer which research questions? What assumptions were therefore made around significance?
RESULTS
- What were the demographics of your focus groups- how do they compare to those of the survey?
- How did you determine that safety was the most important theme? I'm guessing from 'frequency' but this is not clear and talks to the need for detail in the methods section.
- I find that the way that you have chosen to represent your qualitative data are a little confusing- the narrative you have used in the main body of the section talks to sub-themes, but this is not explained.
- What's the relevance of reporting survey demographics?
- What is M/%?
- I'm not sure that you can use the frequency of qualitative statements as your rank- a more valid/reliable way to this would have been to use a ranking question in the interview.
Discussion
- You begin this section with 2 purposes, which aren't made as explicitly clear elsewhere in the manuscript- please make sure that your aims/research questions are clear throughout the sections.
- Why would you assume there would be a male/female difference? Your population are already active!
- At this point I'm still not sure what you mean by trail user motivation, so perhaps this needs better defining earlier.
- You talk about referring patients to locations for PA- how is this relevant to your study, given the population used to develop and 'validate' the tool? I think the narrative around this section (4.1) is a stretch.
- This issue of rurallity comes up again here and in relation to the representitivness of the population used in this study- this is somewhat acknowledged in section 4.3, but it's not really explicit.
- I would like to see some reference in the discussion of Plano and Clark's mixed methods convergant core design and a critique of it's use in your study.
Reviewer 2 Report
See my comments in the attached.
